# Lotus-Flower- and Lotus-Seedpod-Derived Polysaccharide: Structural Characterization and Biological Activity

**DOI:** 10.3390/polym15183828

**Published:** 2023-09-20

**Authors:** Zhiqiang Zhang, Li Wang, Dai Zeng, Xia Ma, Hui Wang

**Affiliations:** 1College of Pharmacy, Henan University of Chinese Medicine, Zhengzhou 450045, China; 2Department of Traditional Chinese Medicine, Henan Agricultural University, Zhengzhou 450002, China; 3College of Animal Medicine, Henan University of Animal Husbandry and Economy, Zhengzhou 450046, China

**Keywords:** extraction, purification, lotus flower, lotus seedpod, polysaccharide, structure, biological activity

## Abstract

Lotus flower polysaccharide (LFP) and lotus seedpod polysaccharide (LSP) were separated by water extract–alcohol precipitation, and their structures and biological activities were investigated. The results of monosaccharide composition showed that LFP and LSP were composed of nine monosaccharides, fucose, rhamnose, arabinose, glucose, galactose, mannose, fructose, galacturonic acid, and glucuronic acid, with the molar percentages of 0.18: 0.43: 2.26: 45.22: 32.14: 4.28: 8.20: 6.28: 1.01 and 2.70: 1.02: 8.15: 45.63: 20.63: 1.44: 2.59: 16.45. LSP and LFP exhibited molecular weights of 9.37 × 10^4^ Da and 1.24 × 10^6^ Da, respectively. SEM showed that LFP and LSP have similar structures; XRD analysis showed that both polysaccharides had crystalline structure and amorphous structure. The results of ABTS+, DPPH, hydroxyl radical scavenging experiment, and a reducing power experiment showed that LFP and LSP had good antioxidant capacity. Cell viability findings showed that polysaccharide concentrations of lotus flower and lotus seedpod could enhance cellular proliferation ranging from 25 to 400 μg/mL without cytotoxicity. By inducing the production of crucial proteins in the TLR4/NF-κB pathway, LFP and LSP were able to induce autophagy in RAW264.7, according to the results of the RT-PCR and Western blotting assays.

## 1. Introduction

*Nelumbo nucifera Gaertn*, also known as a lotus flower and water lily, is a perennial aquatic host plant of the Nymphaeaceae family that is found in large quantities in China, India, Korea, Thailand, and Japan [1]. It has been used for more than 2000 years as an herbal remedy, functional food, and vegetable. The lotus is not only an ornamental plant in China [2] but also an important dual-use resource with rich nutritional value and pharmacological activity [3]. Lotus contains flavonoids, alkaloids, polysaccharides, and other bioactive substances, which have immunomodulatory, antioxidant, memory-improving, and nervous system-regulating effects. The lotus flower is rich in proanthocyanidins, polyphenols, flavonoids, alkaloids, sterols, quercetin, and dietary fiber, which have pharmacological activities such as antioxidant effects, memory and cognitive improvement, antibacterial properties, blood sugar and lipid regulation, anti-tumor effects, and improvement of the nervous system [4].

The modern research indicates that polysaccharide is a kind of natural macromolecule that exists widely in nature [5]. It has many biological activities, such as anti-oxidation, immunomodulation, anti-tumor, and so on [6]. Polysaccharides are widely used in food, health products, cosmetics, and other industries because of their biodegradable, non-toxic properties, biocompatibility, and low processing costs [7]. Researchers are becoming more and more interested in polysaccharides as an essential component of the many components of the lotus. Little research has been performed on how the simultaneous extraction of polysaccharides from various regions of the lotus affects the biological activity of those polysaccharides, compared to the strength of those effects on other portions of the lotus’ polysaccharides. In this paper, the polysaccharides of lotus flower and lotus seedpod were extracted by water extraction and ethanol precipitation, and their molecular weight, monosaccharide composition, and IR properties were assayed; furthermore, the antioxidant and immunomodulatory properties were evaluated by in vitro antioxidant experiments and cell models. The results of this study provide an experimental basis for the research and development of LFP and LSP as pharmaceutical and food supplements.

## 2. Material and Methods

### 2.1. Materials and Reagent

Fresh lotus flower and lotus seedpod were collected from Zhongmou City, Zhengzhou, Henan Province, China. Biotopped Life Science Co., Ltd. (Beijing, China) provided the folinol reagent and coomassie brilliant blue, Sigma Chemical Company (St. Louis, MI, USA) supplied the lipopolysaccharide (LPS), Hanheng Biotechnology Co., Ltd. (Shanghai, China) provided the Cell Counting Kit-8 (CCK-8), and Gibco provided the trypsin and RPMI1640 (Grand Island, NY, USA). GIBCO BRL (Grand Island, NY, USA) provided phosphate buffered saline (PBS) and Vitamin C (Vc). All the above are analytical reagents (AR); 1,1-diphenyl-2-picrylhydrazine radical (DPPH) was acquired from Tokyo Chemical Industry Co., Ltd. (Tokyo, Japan). Other chemicals or reagents are analytically pure.

### 2.2. Extraction and Purification of Lotus-Flower- and Lotus-Seedpod-Derived Poly Nanoparticles

For isolation of lotus-flower- and lotus-seedpod-derived poly nanoparticles, the lotus flower and seedpod were dried overnight in an oven at 45 °C and then powdered and set aside. Then, 1000 g of lotus flower and lotus seedpod powder were weighed into distilled water at a ratio of 1:20 (g:mL), heated in a water bath at 100 °C with stirring for 2 h, and the extracts were extracted twice. The extracts were concentrated properly and then lyophilized after treatment with 80% ethanol alcoholic precipitation, The Sevag method for deproteinization and the H_2_O_2_ method for decolorization were used to obtain crude polysaccharide from the lotus flower and lotus seedpod. Then, 1 g of crude polysaccharide from lotus flower and lotus seedpod were dissolved in pure water. The crude polysaccharide aqueous solution was sequentially centrifuged at 200× *g* for 10 min, 2000× *g* for 20 min, and 10,000× *g* for 30 min to remove large particles and fibers. The final supernatant was ultracentrifuged at 100,000× *g* for 60 min (Beckman Optima XE-100, Beckman, Brea, CA, USA), and the pellets were resuspended in phosphate buffered saline (PBS), transferred to a gradient sucrose solution (15, 30, 45 and 60%) and ultracentrifuged at 150,000× *g* for another 60 min. The band at the 45% sucrose layer was collected and defined as derived nanoparticles according to scanning electron microscope (SEM) examination.

### 2.3. Structure Characterization and Molecular Face

#### 2.3.1. Chemical Analysis of LFP and LSP

The total carbohydrate, total glucuronic acid, total protein, and total phenol contents of two polysaccharides from lotus flower and seedpod were determined by phenol sulfuric acid, m-hydroxybiphenyl, bovine serum protein—Coomassie brilliant blue and forintanol methods, respectively [8,9].

#### 2.3.2. Monosaccharide Composition Analysis and Molecular Weight Assay

The monosaccharide composition of LFP and LSP was determined by electrochemical detector with a Dionex™ CarboPac™ PA20 column on an ICS5000 model ion chromatography system (Thermo Fisher Scientific, Waltham, MA, USA).

DAWN HELEOSII RID-20A differential refractive index detector, shodex OHpak SB-806M HQ, and SB-804M HQ columns (300 × 7.8 mm, shodex, Tokyo, Japan) were used in sequence to evaluate the molecular weight distributions of LFP and LSP.

#### 2.3.3. Infrared Spectrum Analysis

The infrared spectrometer Bruker-Vector 22 of Rheinstetten (Germany) was used. Each sample (1 mg) was mixed with dried KBr (1:100), and the mixture was ground and pressed into granules. The two polysaccharide samples’ infrared absorption spectra were detected in the 400–4000 cm^−1^ frequency range.

#### 2.3.4. X-ray Diffraction (XRD) Analysis

An empyrean X-ray diffractometer (Panalit Co., Ltd., Amsterdam, The Netherlands) was used to obtain X-ray diffraction patterns of LFP and LSP samples at 40 kV and 15 mA. The scanning angle ranges from 2 to 40° with a step size of 0.01 and a scanning speed of 3°/min.

#### 2.3.5. SEM Morphology

The LFP and LSP samples were plated with gold film and fixed on the metal platform, respectively. They were analyzed by JSM-7800 F SEM from Japan, and the images were observed at 3.0 kV voltage and 1000 magnification.

### 2.4. Antioxidant Activity

#### 2.4.1. 2,2′-Azino-bis (3-ethylbenzthiazoline-6-sulfonic acid) (ABTS^+^) Free Radical

Using the Qiu et al. protocol [10], the samples’ capacity to scavenge ABTS^+^ radicals was evaluated. First, the ABTS^+^ working solution was formed by combining 7 mmol/L ABTS^+^ solution with 2.45 mmol/L K_2_S_2_O_8_ solution in a 1:1 (*v*/*v*) ratio and allowing the mixture to react for 16 h at room temperature in the dark. For use, 10 mL of working solution was adjusted with PBS (pH = 7.0) solution until its absorbance at 734 nm was 0.70 ± 0.05. Then, 200 μL of various LFP and LSP sample solution concentrations (0.05, 0.1, 0.5, 1, 2, 4, 6, 8 and 10 mg/mL) were combined with 1.2 mL of the ABTS^+^ solution and thoroughly mixed. The same concentration of vitamin C (Vc) solution was employed as the control group, and the reaction solution was then left at room temperature for 10 min. The absorbance of the reaction solution at 734 nm was measured, and the formula was applied for calculating ABTS+’s ability to scavenge free radicals:Scavenging ability (%) = (1 − A_X_/A_X0_)] × 100
where:

A_x_ represents the sample solution reaction mixture’s absorbance;

A_X0_ indicates the background absorbance of the reaction mixture with the same volume of deionized water PBS mixed solution replacing the ABTS^+^ solution.

#### 2.4.2. DPPH Free Radical

The samples’ ability to scavenge DPPH radicals in vitro was measured using a modified version of the Wang et al. [11]. method. An equivalent volume of newly prepared DPPH (dissolved in ethanol) solution was combined with 500 μL of various concentrations (0.05, 0.1, 0.5, 1, 2, 4, 6, 8, and 10 mg/mL) of LFP and LSP solutions, and the reaction mixture was left to stand in the dark for 30 min at room temperature. As a positive control, the same volume of distilled water served as the blank, and the same concentration of Vc solution served as the positive control. The DPPH radical scavenging ability of the reaction solution was determined using the following formula:Scavenging ability (%) = (1 − A_X_/A_X0_)] × 100
where:

A_X_ represents the sample solution reaction mixture’s absorbance.

A_X0_ denotes the absorbance of an equal volume of anhydrous ethanol replacing the reaction mixture of the DPPH solution.

#### 2.4.3. Hydroxyl Radical

First, 400 μL of polysaccharide sample solutions at various concentrations (0.05, 0.1, 0.5, 1, 2, 4, 6, 8, and 10 mg/mL) were added to 400 μL of 4.5 × 10^−3^ mol/L salicylic acid 70% ethanol solution and 400 μL of 4.5 × 10^−3^ mol/L FeSO_4_ solution, and subsequently, to 800 μL of H_2_O_2_ (6 × 10^−3^ mol/L) solution. The reaction solution was then heated at 37 °C for 30 min with Vc as the positive control and the same amount of distilled water as the blank group in place of the sample solution. The absorbance of the reaction solution at 510 nm was measured, and the ability to scavenge hydroxyl radicals was calculated by the following equation:Scavenging ability (%) = [(A_X_ − A_0_)/(A_X0_ − A_0_)] × 100
where:

A_0_ is the same volume of deionized water used to replace the sample reaction mixture’s absorbance.

A_X_ represents the sample solution reaction mixture’s absorbance.

A_X0_ is the absorbance of the reaction mixture with the same volume of deionized water instead of the H_2_O_2_ solution.

#### 2.4.4. Reducing Power

Modifications were made to the method developed by Gao et al. [12] for measuring the reducing capacity of the samples. Specifically, 500 μL of phosphate buffer solution (pH = 6.8) and 0.1 mL of 1% (*w*/*v*) potassium ferricyanide solution were added to 200 μL of various concentrations (0.05, 0.1, 0.5, 1, 2, 4, 6, 8, and 10 mg/mL) of LFP and LSP sample solutions, and the reaction solution was then incubated at 50 °C for 20 min. After rapidly cooling, the reaction solution was combined with 0.5 mL of 10% (*w*/*v*) trichloroacetic acid solution. For 5 min, the mixture was centrifuged at 8000 rpm. The absorbance of the reaction product was measured at 700 nm using Vc as a positive control. The strength of the sample’s reducing power is represented by the reaction system’s absorbance; the higher the absorbance value, the greater the sample’s reducing power.

### 2.5. Assay of Immunomodulatory Activity of LFP and LSP

#### 2.5.1. Cell Cultivation

The RPMI1640 media was supplemented with 10% fetal bovine serum, 100 U/mL penicillin, and 100 g/mL streptomycin after RAW264.7 were cultured in an incubator at 37 °C and 5% CO_2_. Every two to three days, the cells were passaged.

#### 2.5.2. Cell Viability

The cell viability of SRP was detected by the CCK-8 method [13]. RAW264.7 in the logarithmic growth phase was seeded into 96-well plates at a density of 2 × 10^4^ cells/well, and 100 μL of cell suspension was added to each well. The wells were then grown in a cell incubator at 37 °C and 5% CO_2_ for 24 h to determine the viability of the LFP and LSP. LPS was introduced to the positive control group at a concentration of 1 μg/mL, the administration groups at concentrations of 25, 50, and 50, and the blank control group at a concentration of 100 μL total culture solution. Each well received 10 μL of CCK-8 after 6 h of incubation. After 1 h, the OD value at 450 nm was determined.

#### 2.5.3. Phagocytosis Assay

RAW264.7 in the logarithmic growth phase was inoculated into 96-well plates at a concentration of 2 × 10^4^ cells/well, 100 μL cell suspension was added to each well, and then it was cultured in a cell incubator with 5% CO_2_ at 37 °C for 24 h. The supernatant was absorbed out, washed twice with PBS and 100 μL of neutral red staining solution was added to each well. After incubation in an incubator for 4 h, the supernatant was discarded, the cells were washed twice with PBS, and 200 μL lysis solution (50% ethanol: 50% acetic acid = 1:1) was added to each well. The light absorption was discovered at 540 nm after 4 h of incubation at room temperature.

#### 2.5.4. Scratch-Wound Assay

The method of cell scratch–wound assay was as reported by Zubair et al. [14]. RAW264.7 was inoculated at a density of 1 × 10^6^ cells per well and cultured in 6-well plates for 24 h, then scratched with the tip of a 200 μL gun to ensure uniform damage. The scratched cells were washed three times with PBS to eliminate the damage before being cultured for 24 h in media containing various concentrations of LFP and LSP (25, 50, 100, 200 μg/mL) and LPS (1 μg/mL). Samples were taken, and cell movement in the same place was examined.

#### 2.5.5. Extraction of mRNA and Real-Time Quantitative Polymerase Chain Reaction

Table 1 shows the primer sequences for RT-PCR. The RNA was isolated using the Trizol procedure and evaluated for concentration and quality using Nanodrop8000. The obtained RNA was then used as a template for cDNA synthesis by reverse transcription using HiScriptIIIRTSuperMix qPCR(+gDNA). Fluorescence quantification reaction systems were prepared by Taq Pro Universal SYBR qPCR Master Mix. The 2^−ΔΔCT^ method was used to record the change in the relative expression of each gene at the end of the reaction.

#### 2.5.6. The Western Blotting and Total Protein Extraction Assays

The total protein of cells was separated by protein lysate. The BCA kit was utilized to measure protein quantity, transferred to a PVDF membrane using SDS-PAGE gel electrophoresis, sealed in skim milk for 2 h, and then incubated overnight at 4 °C with diluted primary antibody. The next day, any remaining primary antibody was removed using TBST solution, and the secondary antibody was added and incubated at room temperature for 2 h. After being added, the secondary antibody was incubated for 2 h at room temperature. Three TBST buffer washes were followed by the development of the secondary antibody.

### 2.6. Statistical Analysis

Data analysis was performed with SPSS 19.0 software (SPSS Inc., Chicago, IL, USA). Differences in mean values among the polysaccharide and control groups were analyzed by a one-way analysis of variance (ANOVA). The data were expressed as mean ± standard error. Values of *p* < 0.05 were considered to be statistically significant.

## 3. Results and Analysis

### 3.1. Yield and Chemical Composition

Table 2 displays the findings of the chemical composition of LFP and LSP. The protein level was 4.63 ± 0.07 mg/g and 6.62 ± 0.26 mg/g, the amount of carbohydrates was 22.86% ± 0.52 mg/g and 41.24 ± 0.94 mg/g, and the quantity of polyphenols was 2.89 ± 0.02 mg/g and 6.66 ± 0.07 mg/g, respectively. The polyphenol content was 2.89 ± 0.02 mg/g and 6.66 ± 0.07 mg/g, respectively; and glucuronic acid content was 28.5 ± 0.16 and 34.03 ± 0.35 mg/g, respectively.

### 3.2. Monosaccharide Composition and Molecular Weight

The monosaccharide composition results of LFP and LSP are shown in Table 3. Both are composed of nine monosaccharides, which are glucuronic acid, galacturonic acid, fructose, mannose, galactose, glucose, arabinose, rhamnose, and fucose. The molar percentage of these nine monosaccharides in LFP is 0.18: 0.43: 2.26: 45.22: 32.14: 4.28: 8.20: 6.28: 1.01. The molar percentage of these nine monosaccharides in LSP is 2.70: 1.02: 8.15: 45.63: 20.63: 1.44: 2.59: 16.45: 1.38. The molar percentage of monosaccharide showed that LFP and LSP were mainly composed of glucose and galactose, followed by fructose, which indicated that they might have a similar carbon chain skeleton. The total molar percentages of galacturonic acid and glucuronic acid in LFP and LSP were 7.29% and 17.83%, respectively.

The biological activity of polysaccharides is directly connected to the molecular weight, an important physical and chemical characteristic. Low-molecular-weight polysaccharides have significant biological activity, according to previous research [15,16]. Table 3 displays the molecular weight distributions of LFP and LSP. The weight average molecular weights (Mw) of LFP and LSP were 1.24 × 10^6^ Da and 9.37 × 10^4^ Da, the number average molecular weights (Mn) were 8.62 kDa and 3.18 kDa, and the polymer dispersion indexes PDI of LFP and LSP were 1.50 and 2.95.

### 3.3. Fourier Transform Infrared Spectroscopy Analysis

It is possible to identify the distinctive organic groups of polysaccharides using infrared spectroscopy [17]. FT-IR spectroscopy is a useful method for analyzing the types and vibrations of functional groups in polysaccharides and can infer some possible structural features of their components. FT-IR spectra of LFP and LSP are shown in Figure 1A; no significant differences were observed, indicating that the structures of the two polysaccharides extracted by water extraction were similar. The strong absorption peak at 3400 cm^−1^ indicates O-H stretching vibration [18]. Bands in the 3000–2800 cm^−1^ region represent C-H stretching and bending vibrations, including CH, CH_2_, and CH_3_ [19]. The peak at approximately 2300 cm^−1^ is attributed to the presence of O=C=O [20]. Peaks at approximately 1740 and 1630 cm^−1^ were attributed to C=O stretching vibration and asymmetric stretching, revealing the presence of uronic acid [21]. These peaks, located at 1150, 1079, and 1020 cm^−1^, are mainly determined by the stretching vibration of the C-O-H side group and the C-O-C glycoside vibration of the pyranose ring [22]. The absorption peak of pyranose skeleton symmetric contraction vibration at 620–640 cm^−1^ indicates the presence of the C-O-H and C-O-C structure of a sugar ring, that is, the presence of pyranose in the sample. These results indicate that the same extraction method did not change the types of glycosidic bonds and the main functional groups of polysaccharides from lotus flower and lotus seed.

### 3.4. XRD Analysis

XRD analysis results of LFP and LSP are shown in Figure 1B. In the range of 2θ from 0° to 85°, the X-ray diffraction intensity curves of LFP and LSP were similar. There was only one round peak and a diffraction peak with low peak intensity, and the highest peaks appeared at approximately 2θ of 22.76° and 21.78°, respectively, indicating that LFP and LSP exhibited crystal structure and amorphous structure.

### 3.5. SEM Analysis

To analyze polysaccharide surfaces qualitatively, SEM is employed when conditions for polysaccharide extraction, purification, and preparation may have an impact on the molecules’ shape, activity, and structure [23]. According to the SEM images of LFP and LSP polysaccharides at different magnification in Figure 2, the structures of LFP and LSP are roughly the same. Under the magnification condition of 50 μm, it is obvious that LFP is in the shape of the spherical distribution, the size is different, and the surface is smooth. The LSP surface is smooth and spherical. At 20 μm amplification, the above situation is more prominent. The reason may be that both polysaccharides were extracted by water extraction, and the same extraction method may cause the structure to tend to be similar.

### 3.6. Antioxidant Activity

#### 3.6.1. ABTS^+^ Free Radical Scavenging Activity

The scavenging activity of polysaccharides on ABTS^+^ free radicals is frequently employed in the evaluation of their antioxidant activity. The ABTS^+^ free radical method is an easy way to assess the antioxidant capacity of polysaccharides. Hydrogen atoms or electrons can be supplied by antioxidants to modify the color of ABTS^+^ [24]. As seen in Figure 3A, as the concentration in the measuring range expanded, so did the LFP and LSP ABTS^+^ free radicals’ capacity to scavenge free radicals. LFP’s potent hydrogen supply capacity may be the reason for its remarkable ABTS^+^ free radical scavenging capacity at low concentrations. According to a study, polysaccharides containing uronic acid groups can interact with heterocarbon hydrogen atoms to scavenge ABTS^+^ free radicals. The results demonstrated that LFP possesses effective ABTS^+^ radical scavenging capability.

#### 3.6.2. DPPH Free Radical Scavenging Assay

The simple DPPH free radical assay can be used to assess the polysaccharide’s antioxidant potential. Polysaccharides’ ability to scavenge DPPH free radicals often depends on the antioxidants’ capacity to donate hydrogen [25]. Figure 3B, which used Vc as a positive control, displayed the scavenging activity of DPPH free radicals. The scavenging activity of DPPH free radicals in all samples increased with a concentration within the experimental concentration range. The scavenging ability of LFP and LSP rose, significantly at first and, subsequently, slowly, in the concentration range of 0.05–10.0 mg/mL. At a concentration of 10.0mg/mL, the scavenging rates of LFP and LSP increased to 66.67% and 73.28%, respectively, indicating that LFP and LSP exhibited a clear DPPH free radical scavenging capacity.

#### 3.6.3. Hydroxyl Radical Scavenging Activity

The hydroxyl radical is one of the most active free radicals in the body, and their excess can disrupt the body’s homeostasis. Free radicals can cause significant harm to neighbor biomolecules and potentially induce cell death. Hence, removing hydroxyl radicals is essential [26]. The hydroxyl radical scavenging activity of LSP and LFP are shown in Figure 3C, with Vc as the positive control. The polysaccharides’ capacity to scavenge hydroxyl radicals demonstrated a rising tendency within the experimental concentration range, following a pattern identical to that of Vc. The scavenging capacity of LFP increased rapidly at first, then slowly, while the hydroxyl scavenging capacity of LSP increased slowly. The scavenging rates of LFP and LSP on hydroxyl radicals were 99.13% and 75.43%, respectively, at a concentration of 10.0 mg/mL. According to the findings, LFP and LSP were effective at scavenging hydroxyl radicals.

#### 3.6.4. Reducing Power

Reducing power is correlated with the presence of a reducing agent, which is a significant indicator of natural substances’ antioxidant activity [27]. The decreasing power of LFP and LSP both exhibited an upward trend, as displayed in Figure 3D. The reducing ability of polysaccharides varies with the concentration of the sample. More reducing ends and non-reducing ends are typically present in polysaccharides with lower molecular weights, which enhances their reducing potential. At the concentration of 10.0 mg/mL, the absorbance of LFP and LSP were 1.19 and 1.13, respectively.

### 3.7. Evaluation of LFP and LSP’s Immunomodulatory Activity

#### 3.7.1. Cell Viability

Figure 4A depicts the outcomes of the cell viability test following the administration of LFP and LSP. Different LFP and LSP concentrations had varying effects on the activity of RAW264.7 compared to the control group. RAW264.7 cells can proliferate when exposed to low concentrations of polysaccharides in a concentration-dependent way. When the concentration is ≤400 g/mL, the polysaccharide can promote cell growth, which indicates that the polysaccharide has no toxicity to cells. The concentration ≥600 g/mL can inhibit cell proliferation, indicating that it has certain toxicity to cells.

#### 3.7.2. Effect of LFP and LSP on the Phagocytosis of Macrophage (RAW264.7)

The effect of polysaccharide on phagocytosis was detected by the uptake of neutral red by macrophages [28]. Figure 4B illustrates the calculated findings. Compared to the blank control group, the concentrations of LFP polysaccharide between 25–200 μg/mL and LSP polysaccharide between 25–400 μg/mL enhanced the absorption of neutral red by RAW264.7 substantially (*p* < 0.05).

#### 3.7.3. Effect of LFP and LSP on the Migration of RAW264.7

The most direct way to detect cell migration is the scratch test [29]. As shown in Figure 5, RAW264.7 was treated with LPS (1 μg/mL) and different concentrations of LFP and LSP polysaccharides (25–200 μg/mL), and the percentage of cell scratch–wound on macrophages was observed. According to the Figure 5, after 24 h, the scratch width of each treatment group was clearly less than that of the control group. This clearly promoted the scratch–wound and improved the ability of RAW cells to migrate.

#### 3.7.4. RT-PCR Testing Result

The results of RT-PCR are shown in Figure 6. The results showed that compared with the control group, the mRNA contents of TLR4, MyD88, p65, and Beclin 1 induced by LFP and LSP polysaccharides with a mass concentration of 25–50 μg/mL and LPS on RAW264.7 increased significantly, and the p62mRNA contents decreased significantly (*p* < 0.05).

#### 3.7.5. The Expression of the Autophagy Protein and the TLR4/NF-κB Pathway

Figure 7B–F depicts how LFP and LSP affect the levels of the essential proteins TLR4, MyD88, and NF-κB, as well as the autophagy proteins Beclin 1 and p62 in the TLR4/NF-κB signaling pathway (β-actin serves as an internal control). TLR4 and NF-κB p65 protein expression levels increased in the LFP group when compared to the positive control group, but MyD88, Beclin 1, and p62 levels declined. MyD88 and NF-κB protein expression levels increased in the LSP group, whereas TLR4, Beclin 1, and p62 protein expression levels lowered considerably. TLR4, MyD88, NF-κB p65, and p62 protein expression levels were significantly higher in LFP and LSP polysaccharide groups and LPS groups compared to the blank group, but the relative expression level of Beclin 1 protein was considerably lower. In conclusion, RAW264.7 can undergo autophagy by expressing critical proteins that are activated by LFP and LSP through the TLR4/NF-κB pathway.

## 4. Discussion

The most popular method for extracting polysaccharides from various lotus sections is water extraction [30]. Polysaccharides can be extracted from various portions of the lotus using various methods, including microwave-assisted extraction, ultrasonic-assisted extraction, enzyme-assisted extraction, alkali extraction, and acid extraction [30,31]. As an important component of different parts of the lotus, the polysaccharide has attracted more and more attention from researchers. The effects of simultaneous extraction of polysaccharides from various lotus sections on the bioactivity of lotus leaf polysaccharides have been the subject of numerous studies [32]. However, there are few studies about its effect on the biological activity of polysaccharides in other parts of the lotus. In this study, LFP and LSP were selected for research. They were separated by water extraction and alcohol precipitation, and their yields were 5.57% 0.04%, and 2.51% 0.03%, respectively.

Polysaccharide structure and physiological function are intimately connected [33]. The biological activity of polysaccharides in different regions of the lotus will be influenced by their molecular weight, uronic acid content, protein content, and polyphenol concentration. Feng et al. [34] reported that lotus leaf polysaccharide with low molecular weight and high uronic acid content showed a stronger effect in scavenging ABTS and DPPH free radicals and reducing power of iron ions, which was consistent with the results in this paper that LSP with low molecular weight and high uronic acid content had higher ABTS free radical resistance and reducing power than LFP. Wu et al.’s research also confirmed that lotus leaf polysaccharide with low molecular weight and high uronic acid content has stronger biological activity in immunomodulation in vitro [31]. The six monosaccharides, galactose, glucose, mannose, rhamnose, xylose, and arabinose, made up the majority of the LSP, as reported by Ma Guangqiang et al. [35]. The results of monosaccharide analysis in this paper showed that LFP and LSP are heteropolysaccharides composed of nine monosaccharides with a richer monosaccharide composition.

Herbal polysaccharide can enhance immune activity by regulating cell division and differentiation [36]. Both LFP and LSP can stimulate RAW264.7 to promote cell proliferation, migration, and phagocytosis.

In the control of the immune system and the inflammatory process, NF-κB is crucial [37]. TLR4 is an important member of the toll-like receptor family, which was proven to be involved in mediating immune inflammatory response [38]. Therefore, the TLR4/NF-κB pathway is an important pathway for inflammatory signal transduction. Autophagy is a cellular mechanism of self-degradation, transformation, and energy production. Activation of toll-like receptors in pattern recognition can induce autophagy and regulate the inflammatory response [39]. LC3 is the signature protein of autophagy; P62 is the substrate of autophagy degradation; The identification and packing of degradation substrates are crucial aspects of its function. In mammalian cells, Beclin 1 overexpression encourages autophagy, and Beclin 1 expression rises as autophagy advances. Therefore, LC3, Beclin 1, and p62 are selected as the indexes of autophagy [40]. Our findings indicate that LFP and LSP, at concentrations of 25–50 μg/mL, can dramatically enhance the expression of TLR4, MyD88, NF-κB p65, and p62 proteins. As a result, LFP and LSP may induce autophagy in RAW264.7 by triggering the production of important TLR4/NF-κB pathway proteins.

## 5. Conclusions

In this study, LFP and LSP were extracted from lotus flowers and lotus seedpods by water extraction and alcohol precipitation, The results of monosaccharide composition showed that LFP and LSP were all composed of nine monosaccharides: fucose, rhamnose, arabinose, glucose, galactose, mannose, fructose, galacturonic acid, and glucuronic acid. The molar percentages are 0.18: 0.43: 2.26: 45.22: 32.14: 4.28: 8.20: 6.28: 1.01 and 2.70: 1.02: 8.15: 45.63: 20.63: 1.44: 2.59: 16.45, respectively.

For the polysaccharide structure, FTIR results showed that LFP and LSP had functional groups of polysaccharides; XRD analysis showed that both polysaccharides exhibited crystalline structure and amorphous structure. The crystallinities of LFP and LSP were 22.76° and 21.78°, respectively. SEM showed that the surfaces of LFP and LSP were smooth and spherical.

The biological activities of LFP and LSP were initially investigated. The results of ABTS^+^, DPPH, the hydroxyl radical scavenging experiment, and the reducing power experiment showed that LFP and LSP had good antioxidant capacity. The cell viability data demonstrated that LFP and LSP may enhance cell proliferation and without cytotoxicity. Activating the production of crucial proteins in the TLR4/NF-κB pathway allowed LFP and LSP in the range of 25–50 μg/mL to induce autophagy in RAW264.7, according to the outcomes of RT-PCR and Western blotting assays. These results provide an experimental basis for the research and development of LFP and LSP as pharmaceutical and food supplements.

## Figures and Tables

**Figure 1 polymers-15-03828-f001:**
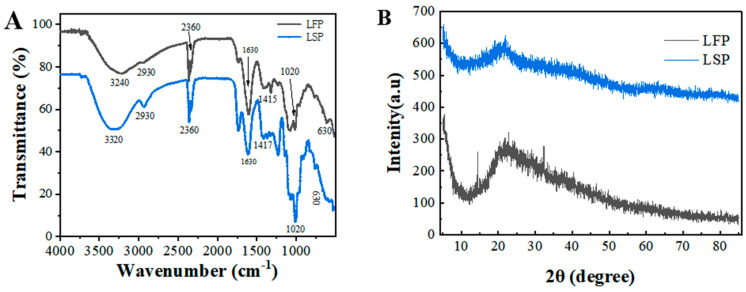
FT-IR spectra (**A**) and XRD (**B**) of LFP and LSP.

**Figure 2 polymers-15-03828-f002:**
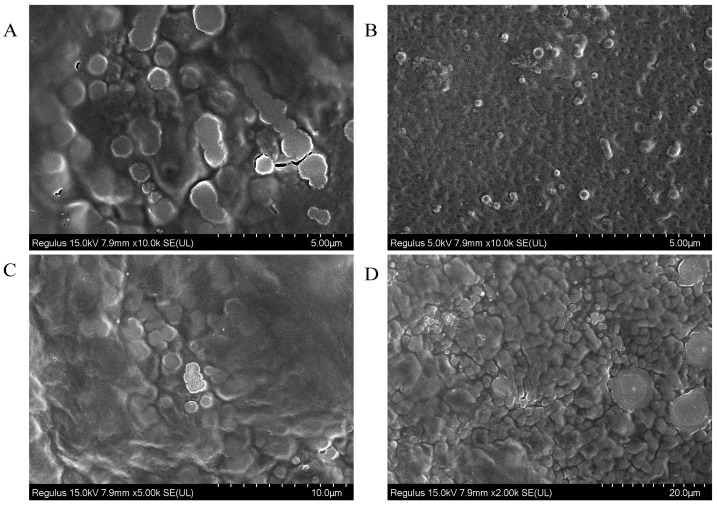
SEM of LFP(**A**,**C**) (magnification 100×) and LSP (**B**,**D**) (magnification 200×), (**A**,**B**) (Scale 5 µm); (**C**) (10 µm); (**D**) (20 µm).

**Figure 3 polymers-15-03828-f003:**
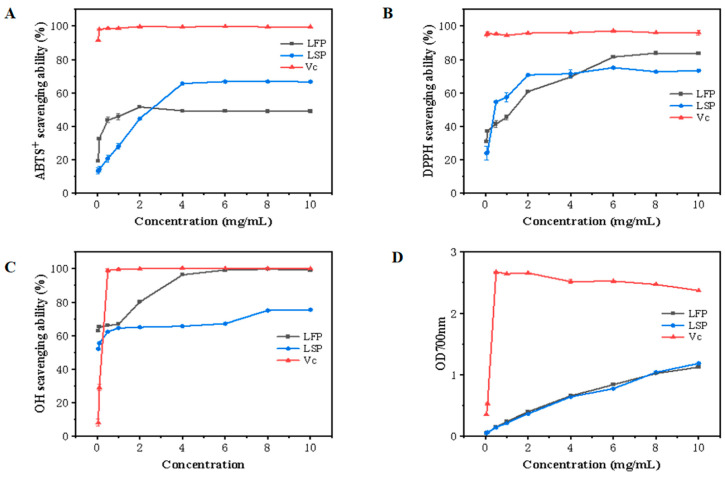
Antioxidant Activity of LFP and LSP. (**A**) demonstrates ABTS^+^ radical scavenging capacity, (**B**) DPPH radical scavenging capacity, (**C**) hydroxyl radical scavenging capability, (**D**) reducing power.

**Figure 4 polymers-15-03828-f004:**
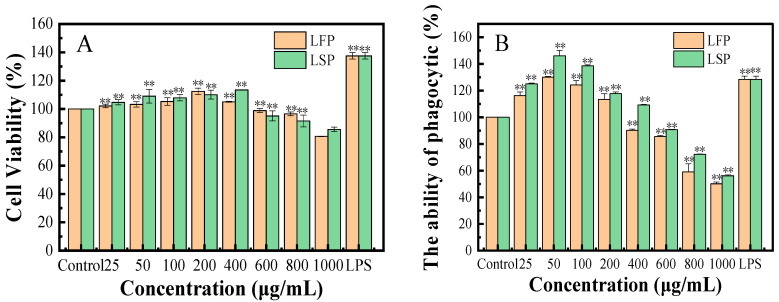
Cell viability and phagocytic activity of LFP and LSP (**A**). Cell vitality; (**B**). Phagocytic activity: ** *p* < 0.01, when compared to a blank control group.

**Figure 5 polymers-15-03828-f005:**
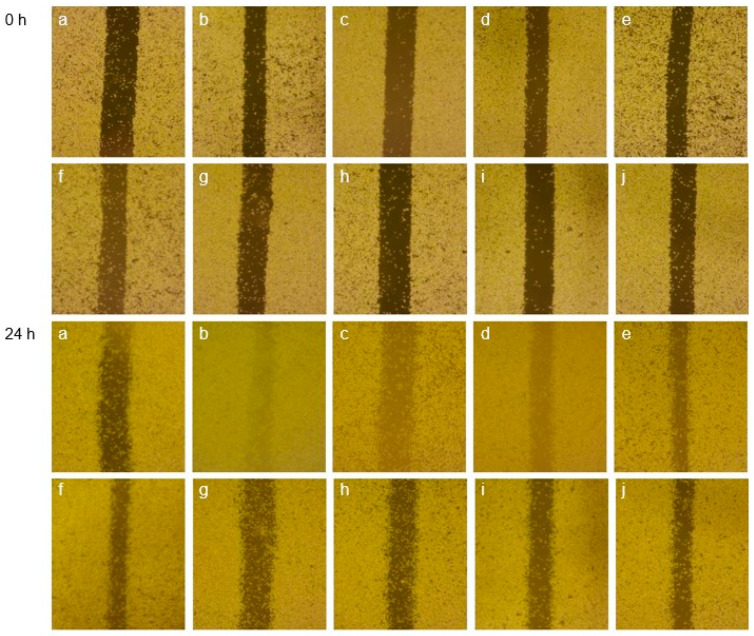
Scratch Test Results of RAW264.7 (**a**–**j**): Control, LPS, LFP (25, 50, 100, 200 μg/mL), LSP (1 μg/mL).

**Figure 6 polymers-15-03828-f006:**
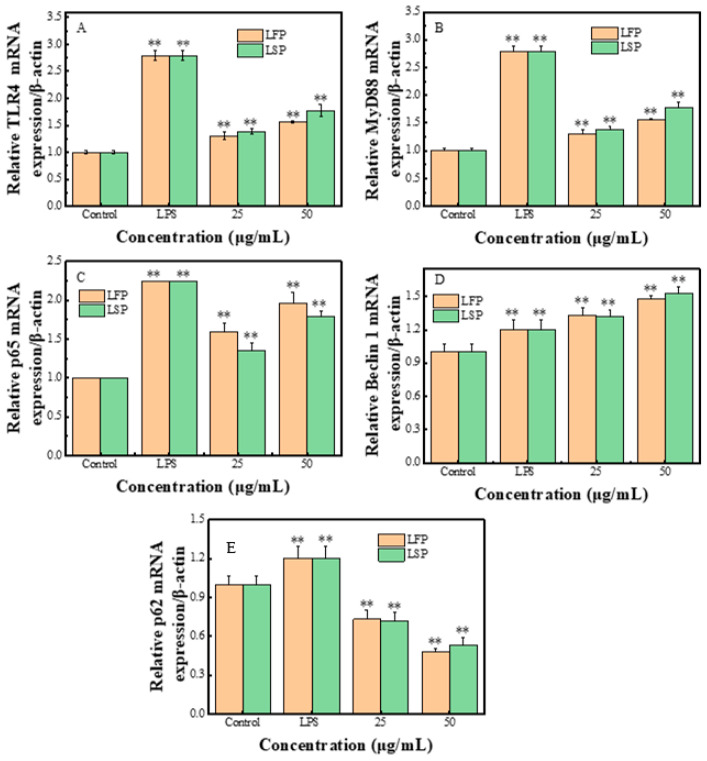
Effects of LFP and LSP on cytokine gene expression in RAW264.7 (**A**–**E**): The expression of TLR4, MyD88, p65, Beclin 1, and p62. ** *p* < 0.01, compared with control.

**Figure 7 polymers-15-03828-f007:**
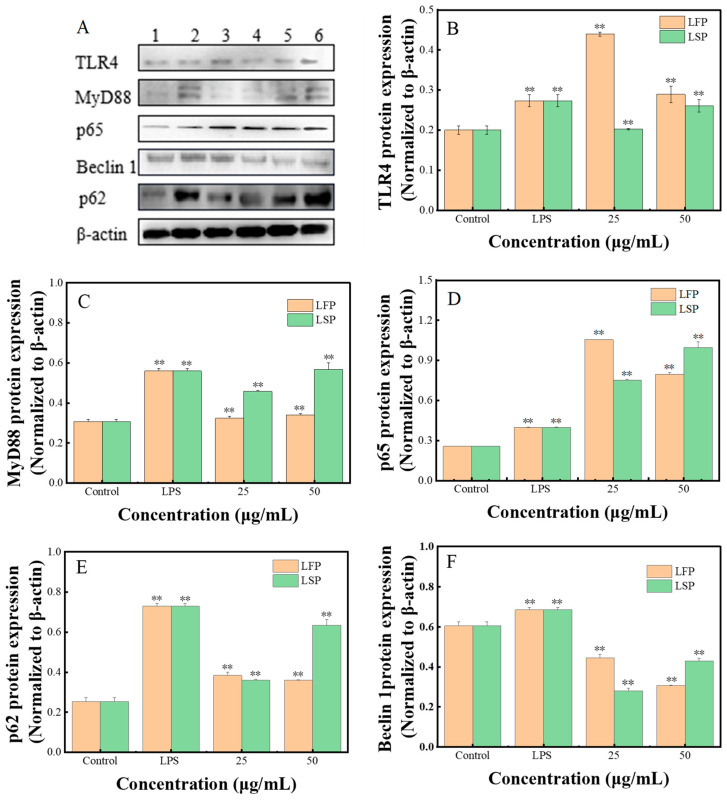
Impacts of LFP and LSP on TLR4/NF-κB Pathway Protein Expression (**A**). Analysis of protein bands by Western blotting; (**B**). TLR4; (**C**). MyD88; (**D**). NF-κB p65; (**E**). Beclin 1; and (**F**). p62 protein expression; ** *p* < 0.01, compared with blank group.

**Table 1 polymers-15-03828-t001:** Lists the RT-PCR primer sequences utilized in this study.

Primer	Forward (5′-3′)	Reverse (5′-3′)
TLR4	ATTTCCGCTTCCTGGTCT	GTCATCCCACTTCCTTCCT
MyD88	CCGCCTGTCTCTGTTCTT	GTCCGCTTGTGTCTCCA
P65	ATGCGCTTCCGCTACAA	GTGACCAGGGAGATGCG
Beclin 1	GAGCGATGGTAGTTCTGGA	CCCGATGCTCTTCACCT
p62	CTTTGACTGAGCGACAGCA	GCCACAACCCCAAACTACA
β-actin	ACCCCAGCAAGGACACTGAGCAAG	GGCCCCTCCTGTTATTATGGGGGT

**Table 2 polymers-15-03828-t002:** Yield and Chemical Composition of LFP and LSP.

Sample	LFP	LSP
Yield (%)	5.57 ± 0.04	2.51 ± 0.03
Carbohydrate (mg/g)	22.86 ± 0.52	41.24 ± 0.94
Protein (mg/g)	4.63 ± 0.07	6.62 ± 0.26
Polyphenol (mg/g)	2.89 ± 0.02	6.66 ± 0.07
Uronic acid (mg/g)	28.5 ± 0.16	34.03 ± 0.35

**Table 3 polymers-15-03828-t003:** Monosaccharide Composition and Molecular Weight Results.

Sample	LFP	LSP
Monosaccharide (mol%)		
Fucose	0.18	2.70
Rhamnose	0.43	1.02
Arabinose	2.26	8.15
Glucose	45.22	45.63
Galactose	32.14	20.63
Mannose	4.28	1.44
Fructose	8.20	2.59
Galacturonic acid	6.28	16.45
Glucuronic acid	1.01	1.38
Molecular weight (Da)		
Polymer dispersity index (PDI)	1.50	2.95
Number—average molecular weight (Mn)	8.23 × 10^5^	3.18 × 10^4^
Weight—average molecular weight (Mw)	1.24 × 10^6^	9.37 × 10^4^

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
