# Peer review of "Lotus-Flower- and Lotus-Seedpod-Derived Polysaccharide: Structural Characterization and Biological Activity"

_polymers, 2023, doi:10.3390/polym15183828_

Round 1

Reviewer 1 Report

I consider the present manuscript “Lotus Flower and Lotus Seedpod derived- polysaccharide nanoparticles: Structural Characterization and Biological Activity” interesting and worthy of attention. Therefore, the manuscript is suggested to be published in the journal after major revisions. Here are my suggestions addressed to the authors:

1. Line 15: remove “the” before “molar percentages…”

2.  Line 17: Please correct “9.37 x 104 Da and 1.24 x 106 Da”

3. Line 20: add space between “that” and “the”

4. Line 31: species name should be in italic: “Nelumbo nucifera

5. Lines 55, 131, and 408: the word “in vitro” should be in italic

6. Lines 68, 173, 204, 212, 334, 346, 352, and 370: sometimes the authors capitalized the first letter of words and sometimes not. All should be the same, please check all sections, subsections and subsubsections and make them as required.

7. Line 59: “2.1. Materials and Reagents” in place of “Reagent

8. Lines 69 and 76: the words “Lotus Flower and Lotus Seedpod” in the text should be minuscule.

9. Lines 76-77: the sentences “Lotus Flower and Lotus Seedpod” and Lotus Flower and Lotus Seedpod were dissolve in pure water” should not be in italic. and "were dissolved” in place of “were dissolve”

10. Line 80: before using the abbreviation “PBS”, it should first write the name completely.  The same for “GDNPs” in line 83, “XRD” in line 102,

11. I suggest more describing the subsection “2.3.1. Chemical Analysis of LFP and LSP” and “2.3.3. Monosaccharide Composition Analysis and Molecular Weight Assay”, to well understand the methodology and conditions used.

12. Line 112: please correct “protocol”, the first letter has to be minuscule.

13. Line 119: what that means “Vc”???, please write in the first one the name completely then uses the abbreviation.  Please check all the abbreviations, first write the names completely with abbreviations in brackets, then use abbreviations alone.

14. Lines 125, 141, 157: The authors explain the “A0” but it does not appear in the three equations of lines 123, 139 and 155 respectively. Any explanation???

15. Line 201: please correct “LFP” in place of “lFP”

16. Line 221: The subsection on the description of data analysis/statistical analysis is missing; please add this part.

17. PBS, the control Vc… and all reagents and standards used should be added to the subsection of “2.1. Reagent

18. Results section: I suggest adding a statistical comparison between the results of LFP and LSP, especially in Tables 2 and 3 to show significant differences between the two samples tested. How many replicates have you made in your chemical experiments?

19. Line 269: Please correct Figure 1 not 2

20. Lines 276: Figures 2 - I suggest adding a legend to the chart description for better clarity.

21. Line 287: Antioxidant Activity subsection – I suggest determining the EC50 or IC50 values (The EC50 is the concentration of a drug that gives the half-maximal response. The IC50 is the concentration of an inhibitor where the response (or binding) is reduced by half) of each sample and antioxidant assay to improve well this subsection.

22. Line 289: It is Figure 3, not 5. Please correct it and I suggest adding a legend to the chart description for better clarity.

23. Line 290: Figure 3, not 5 and should not be in bold.

24. Lines 297, 306, 319 and 329: Please correct “Figure 3” not “Figure 5”

25. Line 332: please leave the space between the number and unit “10.0 mg/mL”

26.  Lines 356 and 366: Please write the full word “Figure 5” and “Figure 6” not Fig.

27.  Discussion section: Some results are not well discussed by comparing them with other findings.

28. Conclusion section: This section is a repetition of the abstract. It needs to be rewritten and improved

29.  Author Contributions section is missing., please add it

Author Response

Reviewer1:I consider the present manuscript “Lotus Flower and Lotus Seedpod derived- polysaccharide nanoparticles: Structural Characterization and Biological Activity” interesting and worthy of attention. Therefore, the manuscript is suggested to be published in the journal after major revisions. Here are my suggestions addressed to the authors:

  1. Line 15: remove “the” before “molar percentages…”

“the” has been removed.

  1. Line 17: Please correct “9.37 x 104Da and 1.24 x 106 Da”

 “9.37 x 104 Da and 1.24 x 106 Da” has been changed into” 93.7 KDa and 20.7 KDa”,

  1. Line 20: add space between “that” and “the”

   Space had been added.

  1. Line 31: species name should be in italic: “Nelumbo nucifera

   “Nelumbo nucifera” had been changed to in italic.

  1. Lines 55, 131, and 408: the word “in vitro”should be in italic

 “in vitro”  had been changed to in italic.

  1. Lines 68, 173, 204, 212, 334, 346, 352, and 370: sometimes the authors capitalized the first letter of words and sometimes not. All should be the same, please check all sections, subsections and subsubsections and make them as required.

   The content has been amended.

  1. Line 59: “2.1. Materials and Reagents” in place of “Reagent

Reagent ”has been changed into ”Materials and Reagents”  

  1. Lines 69 and 76: the words “Lotus Flower and Lotus Seedpod” in the text should be minuscule. 

    The content has been amended.

  1. Lines 76-77: the sentences “Lotus Flower and Lotus Seedpod” and Lotus Flower and Lotus Seedpod were dissolve in pure water” should not be in italic. and "were dissolved” in place of “were dissolve”

The content has been amended.

  1. Line 80: before using the abbreviation “PBS”, it should first write the name completely.  The same for “GDNPs” in line 83, “XRD” in line 102,

The completely name of all the abbreviation had been added.

  1. I suggest more describing the subsection “2.3.1. Chemical Analysis of LFP and LSP” and “2.3.3. Monosaccharide Composition Analysis and Molecular Weight Assay”, to well understand the methodology and conditions used.
  2. Line 112: please correct “protocol”, the first letter has to be minuscule.

The first letter of “protocol” has to be changed into minuscule.

  1. Line 119: what that means “Vc”???, please write in the first one the name completely then uses the abbreviation.  Please check all the abbreviations, first write the names completely with abbreviations in brackets, then use abbreviations alone.

The completely name of all the abbreviation had been added.

  1. Lines 125, 141, 157: The authors explain the “A0” but it does not appear in the three equations of lines 123, 139 and 155 respectively. Any explanation???

    The three formulas have been modified

  1. Line 201: please correct “LFP” in place of “lFP”

 “IFP” has been changed into” LFP”,

  1. Line 221: The subsection on the description of data analysis/statistical analysis is missing; please add this part.

    “statistical analysis” had been added.

  1. PBS, the control Vc… and all reagents and standards used should be added to the subsection of “1. Reagent

     Reagent source of PBS and Vc were added.

  1. Results section: I suggest adding a statistical comparison between the results of LFP and LSP, especially in Tables 2 and 3 to show significant differences between the two samples tested. How many replicates have you made in your chemical experiments?

  1. Line 269: Please correct Figure 1not 2

“Figure 2B” has been changed into “Figure 1B”

  1. Lines 276: Figures 2 - I suggest adding a legend to the chart description for better clarity.
  2. Line 287: Antioxidant Activitysubsection – I suggest determining the EC50 or IC50 values (The EC50 is the concentration of a drug that gives the half-maximal response. The IC50 is the concentration of an inhibitor where the response (or binding) is reduced by half) of each sample and antioxidant assay to improve well this subsection.
  3. Line 289: It is Figure 3, not 5. Please correct it and I suggest adding a legend to the chart description for better clarity.

“Figure 5” has been changed into “Figure 3”

  1. Line 290: Figure 3, not 5 and should not be in bold.

“Figure 5” has been changed into “Figure 3”

  1. Lines 297, 306, 319 and 329: Please correct “Figure 3” not “Figure 5”

“Figure 5” has been changed into “Figure 3”

  1. Line 332: please leave the space between the number and unit “10.0 mg/mL”

    The space between the number and unit has been added

  1. Lines 356 and 366: Please write the full word “Figure 5” and “Figure 6” not Fig.

    The content has been amended.

  1. Discussion section: Some results are not well discussed by comparing them with other findings.

   The results have been rewrite.

  1. Conclusion section: This section is a repetition of the abstract. It needs to be rewritten and improved

   This section have been rewritten.

  1. Author Contributions section is missing., please add it

Author Contributions section has been added.

Reviewer 2 Report

The authors did interesting research on polysaccharide nano-particles extracted from LFP and LSP.

However, I would like to suggest a major revision of this manuscript.

1- The discussion of this manuscript is unclear. The information of each method has a weak connection with the explanation of the result.

2- The purpose of this study was not clearly mentioned. The authors must improve the structure of the manuscript.

3- Why the author used XRD on polysaccharides while their peaks are unclear?

Additional information on XPS may be included to improve this study.

4- SEM images are unclear. Please use better images and add TEM for better information if it is possible.

5- What is the meaning of the yield of compounds obtained from table 2? Please provide more information to explain.

Author Response

Reviewer2:The authors did interesting research on polysaccharide nano-particles extracted from LFP and LSP.

However, I would like to suggest a major revision of this manuscript.

  • The discussion of this manuscript is unclear. The information of each method has a weak connection with the explanation of the result.

  This discussion of this manuscript have been rewrite.

  • The purpose of this study was not clearly mentioned. The authors must improve the structure of the manuscript.

   I have been rewritten this manuscript.

3- Why the author used XRD on polysaccharides while their peaks are unclear?

Additional information on XPS may be included to improve this study.

 Referring to previous published articles, XRD was used to characterize the structure of polysaccharides

  • SEM images are unclear. Please use better images and add TEM for better information if it is possible.

  Due to limited funding, we used SEM to focus on the structure of polysaccharides, consistent with the previously published literature.

Reviewer 3 Report

Overall, the authors presented nanoparticles-liked lotus-derived polysaccharide nanoparticles. The nanoparticles were isolated from Lotus flower (LFP) and lotus seedpod (LSP) respectively. The structures and biological activities were investigated using different techniques. FTIR results showed that LFP and LSP had functional groups of polysaccharides; XRD analysis showed they showed both polysaccharides crystalline structure and amorphous structure. SEM showed that the surface of LFP and LSP were smooth and spherical. scavenging experiment for ABTS+, DPPH, hydroxyl radical were also conducted to show the reducing power. The results showed that LFP and LSP had good antioxidant capacity. Cell viability was also carried out to study the toxicity of the particles. The results showed that polysaccharide concentrations of lotus flower and lotus seedpod could enhance cellular proliferation ranging from 25 to 400 μg/mL without cytotoxicity. In general, the author formulated a good story to separate useful polysaccharides from lotus flower and lotus seedpod and show their potential medical applications. However, the paper has many issues in terms of formatting and experimental design.

Comment 1: Writings need to be rechecked for the whole paper, especially on the material and methods parts. Fonts are not unified and need to be corrected. Somewhere the author used Italic, while somewhere was not.

Comment 2. Figure 2, the SEM images need to be detailed for each pic. A, B, C and D are not described in the caption. Also, the morphology, shape and details of the nanoparticles from LFP and LSP are not well presented and shown in the SEM images. SEM images need to be further zoomed in to show each particle, then the results can be compared between LFP and LSP. Same for the writing of SEM section, the whole paragraph needs to be examined. Major revision is needed.

Comment 3. For the data in Figure 5, why some of the plots have error bars and some of the plots do not have it? Also, Vc does not have a full name in the paper. Can you elucidate a bit more for the for of Vc as the positive control, why chose to use Vc. Besides using Vc as a control, how the natural decomposition of the radicals is excluded? Should there be a blank control also? Major revision is needed.

Comment 4: The Figures’ number is not well organized. The number jumped from 2 to 5 and there are two Figure 5.

Comment 5: Same for the scratch test results, the Figure captions are lacking of detailed descriptions. There are two sets of pics from a-j that are not differentiated and described. The writing part is also lacking detailed discussion.

Author Response

Reviewer3Overall, the authors presented nanoparticles-liked lotus-derived polysaccharide nanoparticles. The nanoparticles were isolated from Lotus flower (LFP) and lotus seedpod (LSP) respectively. The structures and biological activities were investigated using different techniques. FTIR results showed that LFP and LSP had functional groups of polysaccharides; XRD analysis showed they showed both polysaccharides crystalline structure and amorphous structure. SEM showed that the surface of LFP and LSP were smooth and spherical. scavenging experiment for ABTS+, DPPH, hydroxyl radical were also conducted to show the reducing power. The results showed that LFP and LSP had good antioxidant capacity. Cell viability was also carried out to study the toxicity of the particles. The results showed that polysaccharide concentrations of lotus flower and lotus seedpod could enhance cellular proliferation ranging from 25 to 400 μg/mL without cytotoxicity. In general, the author formulated a good story to separate useful polysaccharides from lotus flower and lotus seedpod and show their potential medical applications. However, the paper has many issues in terms of formatting and experimental design.

Comment 1: Writings need to be rechecked for the whole paper, especially on the material and methods parts. Fonts are not unified and need to be corrected. Somewhere the author used Italic, while somewhere was not.

    The problem of italics and letter capitalization has been checked and corrected.

Comment 2. Figure 2, the SEM images need to be detailed for each pic. A, B, C and D are not described in the caption. Also, the morphology, shape and details of the nanoparticles from LFP and LSP are not well presented and shown in the SEM images. SEM images need to be further zoomed in to show each particle, then the results can be compared between LFP and LSP. Same for the writing of SEM section, the whole paragraph needs to be examined. Major revision is needed.

  This section have been rewritten.

Comment 3. For the data in Figure 5, why some of the plots have error bars and some of the plots do not have it? Also, Vc does not have a full name in the paper. Can you elucidate a bit more for the for of Vc as the positive control, why chose to use Vc. Besides using Vc as a control, how the natural decomposition of the radicals is excluded? Should there be a blank control also? Major revision is needed.

 This section have been revision.

Comment 4: The Figures’ number is not well organized. The number jumped from 2 to 5 and there are two Figure 5.

 The Figures’ number has been checked and modified

Comment 5: Same for the scratch test results, the Figure captions are lacking of detailed descriptions. There are two sets of pics from a-j that are not differentiated and described. The writing part is also lacking detailed discussion.

 This section have been rewritten.

Round 2

Reviewer 1 Report

The revised manuscript is much better after major revision. However, it still needed some minor revisions before final acceptance.

1.     Line 17: Please correct “9.37 x 104 Da and 1.24 x 106 Da”, (10 to the 4th power and 10 to the 6th power)

2.     Line 29: “Nelumbo nucifera” stills appearing not italic.

3.     Lines 53 and 532: still “in vitro” not italic.

Author Response

Thank you very much for the experts' comments on the article!

Q1: I have checked the error and corrected it in Line 17.

Q2: I have rewritten it!

Q3: I have rewritten it!

Reviewer 2 Report

The authors did some minor changes in this version. However, they need to do minor revision before I can accept this manuscript.

1/ Though the explanation has minor changes, the images of SEM in Figure 2 remain unclear, and their caption does not have an appropriate explanation.

Author Response

Thank you very much for the expert's advice

I have upload new High definition picture of SEM .

Reviewer 3 Report

The manuscript looks fine after major revision. English spelling and formating need to be rechecked for better quality.

Author Response

Thanks very much for the experts' opinions. The grammar of the whole article has been re-checked and modified